# Alloplastic Bone Substitutes for Periodontal and Bone Regeneration in Dentistry: Current Status and Prospects

**DOI:** 10.3390/ma14051096

**Published:** 2021-02-26

**Authors:** Shunsuke Fukuba, Munehiro Okada, Kohei Nohara, Takanori Iwata

**Affiliations:** Department of Periodontology, Graduate School of Medical and Dental Sciences, Tokyo Medical and Dental University, Tokyo 113-8549, Japan; sfukuba.peri@tmd.ac.jp (S.F.); mokada.peri@tmd.ac.jp (M.O.); nohara.peri@tmd.ac.jp (K.N.)

**Keywords:** alloplastic bone substitutes, bone graft materials, guided bone regeneration, periodontal regeneration, peri-implantitis, synthetic graft

## Abstract

Various bone graft products are commercially available worldwide. However, there is no clear consensus regarding the appropriate bone graft products in different clinical situations. This review is intended to summarize bone graft products, especially alloplastic bone substitutes that are available in multiple countries. It also provides dental clinicians with detailed and accurate information concerning these products. Furthermore, it discusses the prospects of alloplastic bone substitutes based on an analysis of the current market status, as well as a comparison of trends among countries. In this review, we focus on alloplastic bone substitutes approved in the United States, Japan, and Korea for use in periodontal and bone regeneration. According to the Food and Drug Administration database, 87 alloplastic bone graft products have been approved in the United States since 1996. According to the Pharmaceuticals and Medical Devices Agency database, 10 alloplastic bone graft products have been approved in Japan since 2004. According to the Ministry of Health and Welfare database, 36 alloplastic bone graft products have been approved in Korea since 1980. The approved products are mainly hydroxyapatite, β-tricalcium phosphate, and biphasic calcium phosphate. The formulations of the products differed among countries. The development of new alloplastic bone products has been remarkable. In the near future, alloplastic bone substitutes with safety and standardized quality may be the first choice instead of autologous bone; they may offer new osteoconductive and osteoinductive products with easier handling form and an adequate resorption rate, which can be used with growth factors and/or cell transplantation. Careful selection of alloplastic bone graft products is necessary to achieve predictable outcomes according to each clinical situation.

## 1. Introduction

Dental bone graft materials have been commonly used with growth factors and/or barrier membranes in situations such as periodontal regeneration therapies and guided bone regeneration procedures before implant placements [1,2]. In recent years, these materials have also been applied to bone defects caused by peri-implantitis [3,4]. In the early 20th century, autologous bone from intraoral and extraoral sites was commonly used for periodontal and bone regeneration [5,6,7]. Autologous bone is considered the gold standard because it is the only bone graft that has the following three properties: osteogenesis, osteoinduction, and osteoconduction [8,9,10]. Osteogenesis is a property of autologous grafts, whereby new bone is formed by osteoblast cells derived from the graft. Osteoinduction is a property shared among autologous and allogeneic grafts, as well as intrinsic bone matrix proteins (e.g., bone morphogenetic proteins), that involves host stem cell differentiation into osteoblastic cells. Osteoconduction is a mechanical structure property comprising biocompatibility for the migration of osteogenic cells [11,12].

While periodontal and bone regeneration therapies using autologous bone have achieved predictable clinical outcomes, the harvesting of autologous bone graft requires a secondary surgical site (i.e., a donor site) and increases postoperative patient discomfort [13]. To address these problems, alternatives to autologous bone graft materials have been developed. The main advantages of using bone graft substitutes are unlimited availability and reduced morbidity [14,15]. Bone graft substitutes possess structural characteristics and/or chemical compositions similar to those of natural osseous tissue; accordingly, implantation of these substitutes promotes bone formation. Ideally, bone substitutes should be biocompatible (i.e., able to interface with the organism without eliciting an adverse response), osteoinductive, osteoconductive, absorbable (i.e., eventually be completely replaced by host tissues), safe, easy to use, and cost-effective [16]. There are several categories of dental bone graft substitutes such as allogeneic bone, xenogenic bone, and alloplastic materials; each has unique properties. 

Allogeneic bone grafts are obtained from different individuals of the same species; these have full osteoconduction and partial osteoinduction capabilities [17]. Allogeneic bone grafts have been widely used and constitute an attractive alternative to autologous bone. Allogeneic bone grafts do not require a donor site or abundant supply, but exhibit variable regenerative abilities due to the absence of information concerning donor conditions (e.g., age and systemic health); they also may carry unknown infectious agents and are the focus of ethical and religious controversies [18,19]. In contrast, xenogenic bone grafts are obtained from different species, typically cattle or pigs; these only possess osteoconduction capability [20]. Xenogenic bone grafts have advantages similar to those of allogeneic bone grafts. However, xenogenic bone grafts carry a risk of infectious disease transmission (e.g., bovine spongiform encephalopathy and Creutzfeldt–Jakob disease); they are also the focus of ethical and religious controversies [21]. Finally, alloplastic bone substitutes are synthetic materials that contain some of the essential chemical components of natural bone (e.g., calcium and phosphate) and are known to promote bone regeneration, although they do not necessarily resemble its natural structure [22,23]. Common advantages of alloplastic bone substitutes are the standardized product quality and absence of infectious disease risk, compared with allogeneic and xenogenic bone grafts [24,25]. Since the regenerative abilities of alloplastic bone substitutes are weak, they are often applied with growth factors and/or membranes [22,25]. The main advantages of alloplastic bone substitutes involve their biological stability and volume maintenance that allow cell infiltration and remodeling [25]. Alloplastic bone substitutes have altered osteoconductive capabilities that depend on their compositions and manufacturing methods, as well as their mechanical properties, crystal structures, pore sizes, porosities, and absorption rates [26,27,28]. 

In this review, we focus on alloplastic bone substitutes. Most alloplastic bone substitute products have been approved by the United States Food and Drug Administration (FDA), as well as the Japanese Pharmaceuticals and Medical Devices Agency (PMDA), and the Korean Ministry of Health and Welfare (MOHW) for periodontal and oral implant applications. The biocompatibility, safety, and efficacy of these substitutes in periodontal regeneration and guided bone regeneration (GBR) have been established in various preclinical and clinical studies. However, the levels of evidence vary among substitutes [22,24,25]. There are many alloplastic bone substitutes available for dentistry-related applications worldwide. Each of these products has inherent characteristics that may influence clinical outcomes; this may create difficulty for dentists and other clinicians involved in oral health to select the most appropriate alloplastic bone substitute products for particular clinical situations. There is no clear consensus concerning the selection of appropriate alloplastic bone substitutes for periodontal or implant indications. Each clinician primarily uses specific products in accordance with their personal preferences; however, it is difficult to determine whether the product certification or quality is properly managed. 

Through a comprehensive survey of the available alloplastic bone substitutes, this review aimed to help dentists and other clinicians involved in oral health to select the appropriate products in accordance with specific patient indications and considering the properties of each product. This review also assessed trends in terms of their alloplastic material formulations and clarified differences in products available among the countries surveyed.

## 2. Methodology

Thorough research was conducted by the authors using the terms “alloplastic”, “synthetic”, “bone graft”, and “substitute” in multiple databases to identify alloplastic bone substitute products approved by United States FDA, Japanese PMDA, and Korean MOHW for periodontal and oral implant applications, using an approach described in previous reports [29,30,31,32]. The FDA 510(k) website is a formal database of 510(k) applications and FDA decisions on those applications that have been active since 1996 [31]. The PMDA website is also a formal database of medical devices approved since 2004 [30]. The MOHW website is a formal database of medical devices approved since 1980, as well as information such as their applications and available forms [32]. Other websites were also visited in an exhaustive search for the following product information: commercial names, manufacturers, compositions, indications, and approval dates [33]. Information available on those other websites is publicly accessible through the FDA, PMDA, and MOHW databases. Alloplastic bone substitutes that were not marketed or listed on the aforementioned websites in the United States, Japan, and Korea were not included in this review. Growth factors, cell-based bone repair therapeutics, allogeneic bone grafts, xenogenic bone grafts, and cellular bone matrices for other medical fields (e.g., orthopedic applications) were excluded because those products were not within the scope of this review. Bone graft products that combined alloplastic bone substitutes with allogeneic and/or xenogenic bone grafts were also excluded. After the exhaustive identification and selection of alloplastic bone graft products, the official websites of all included products were visited to obtain information regarding the following details for each alloplastic bone graft product: available form, particle size or block dimensions, porosity, compressive strength, resorption rate, and volume/weight options. When no information was available, the missing data were categorized as undisclosed.

## 3. Results

### 3.1. Alloplastic Bone Graft Products Approved by the United States FDA

The regulatory authority for medical devices in the United States is the FDA. The Center for Drug Evaluation and Research oversees reviews of drug registration applications. In total, 87 alloplastic bone graft products were approved by the FDA from 1996 to December 2020: 15 hydroxyapatite (HA), 21 β-tricalcium phosphate (β-TCP), 18 biphasic calcium phosphate (BCP), 5 calcium sulfate (CS), 5 calcium phosphate (CP: detailed composition was not confirmed), 11 bioglass (BG), and 4 others (e.g., carbonate apatite). Information regarding commercial names, manufacturers, compositions, indications, available forms, morphologies (e.g., particle size or block dimensions and volume/weight options), and approved dates is shown in Table 1 (in alphabetical order according to company name). Of all FDA-approved products, β-TCP was the most common at 23%, followed by BCP at 22% and HA at 18% (Figure 1). In terms of clinical indications, 78% were indicated for periodontal defects; 34% were indicated for the treatment of the furcation defect. Of all approved products, 70% had indications for ridge augmentation, GBR, ridge preservation, and sinus lift prior to implant placement; 20% had indications for peri-implantitis. Most alloplastic bone graft products were available in particulate form (79.7%), followed by putty (10.1%), paste (5.1%), gel (3.8%), and plaster (1.3%) (Figure 1). The distribution of indications according to product components is shown in Figure 2. The most common alloplastic bone graft products approved between 2000 and 2009 were β-TCP (26%), HA (22%), BCP (19%), and BG (16%); among products approved between 2010 and 2020, BCP was the most common (33%), followed by β-TCP (22%) and HA (13%).

### 3.2. Alloplastic Bone Graft Products Approved by the Japanese PMDA

The regulatory authority for medical devices in Japan is the Japanese Ministry of Health, Labor and Welfare. The PMDA oversees reviews of drug registration applications. In total, 10 alloplastic bone graft products were approved by the PMDA from 2004 to December 2020: three HA, four β-TCP, one BCP, one carbonate apatite, and one octacalcium phosphate. There were no approved products consisting of CS or BG for periodontal or oral implant applications. Information regarding commercial names, manufacturers, compositions, indications, available forms, morphologies (e.g., porosities, particle sizes, compressive strengths, and resorption rates), indications, and approved dates is shown in Table 2 (in alphabetical order according to composition). HA and β-TCP comprised 80% of all alloplastic bone graft products (Figure 1). Notably, 70% of alloplastic bone graft products were indicated for periodontal defects, compared with 40% for GBR before implant placement. No approved products were indicated for peri-implantitis (Figure 2). Although most products were available in particulate form, sponge, disc, or rod form products have been approved in recent years (Figure 1).

### 3.3. Alloplastic Bone Graft Products Approved by the Korean MOHW

The Korean MOHW is the regulatory authority for the safety and efficacy of medical devices in Korea. All medical devices are commercialized after they have been priced, classified, and registered by the Health Insurance Review and Assessment Service. Post-application management is required for the manufacturer (or representative importer) to receive a certificate of Good Manufacturing Practice from the Ministry of Food and Drug Safety. In total, 36 alloplastic bone graft products have been approved by the Health Insurance Review and Assessment Service from 1980 to December 2020: 4 HA, 8 β-TCP, 15 BCP, and 1 CP (detailed composition was not confirmed). Information regarding commercial names, manufacturers, compositions, indications, available forms, and morphologies is shown in Table 3. Of all MOHW-approved products, BCP was the most common at 54%, followed by β-TCP at 29% and HA at 14% (Figure 1). In terms of clinical indications, 42% were indicated for periodontal defects; 58% had indications for ridge augmentation, GBR, ridge preservation, and sinus lift prior to implant placement. Two products were indicated for peri-implantitis. Most alloplastic bone graft products were available in particulate form (82%), followed by injection (11%), plug (4%), and block (4%) (Figure 1).

## 4. Discussion

There have been multiple reports on bone graft products available in single countries [31,32]. However, to the best of our knowledge, this is the first review to comprehensively analyze all alloplastic bone graft products available in multiple countries, including the United States, Japan, and Korea. It is also the first study to compare the distributions of these products among various countries.

### 4.1. Properties and Synthetic Routes of Each Composition of Alloplastic Bone Substitutes

The properties of alloplastic bone substitutes are known to vary according to their compositions, as follows. 

CP is a generic term that loosely describes various compositions. LeGeros has described the following types of commercially available CP compounds: (1) calcium HA: Ca_10_(PO_4_)_6_(OH)_2_, either naturally derived (coralline or bovine) or synthetic; (2) β-TCP: Ca_3_(PO_4_)_2_; (3) BCP, consisting of a mixture of β-TCP and HA; and (4) unsintered CPs [34,35]. 

Pure HA (Ca_10_(PO_4_)_6_(OH)_2_) is among the least soluble of the CP compounds and is not found in biologic systems [36]. Synthetic HA is prepared by numerous techniques, broadly divided into (1) solid-state chemical reactions or (2) wet reactions. These preparations have distinct sintering temperatures.

β-TCP (β-Ca_3_[PO_4_]_2_) is one of the two polymorphs of TCP. Typically, β-TCP is prepared by sintering calcium-deficient HA to high temperatures [36]. It can be also be prepared at lower temperatures in water-free mediums or by solid-state acid–base chemical interactions.

Bioactive glasses (BGs) are amorphous materials, based on acid oxides (e.g., phosphorus pentoxide), silica (or alumina oxide), and alkaline oxides (e.g., calcium oxide, magnesium oxide, and zinc oxide). BGs possess an interconnective pore system and are available in both compact and porous forms [37]. The bioactivity of the BG surface enables the growth of osseous tissue [38].

CS is the oldest ceramic bone substitute material, first described by Dressman in 1892 for the filling of osseous defects in human patients [39]. Recent studies continue to demonstrate the bone healing properties of CS [40,41]. CS hemihydrate (CaSO_4_·1/2H_2_O) powder is hydrated to form CS dihydrate (CaSO_4_·2H_2_O), undergoing a slight exothermic reaction to set to a solid form. 

The resorption rate of bone grafts is a feature that clinicians consider very important; there is substantial variability among alloplastic materials. HA is known to require a long interval for replacement by native bone due to its low substitution rate [42]. If socket grafting and early re-entry for implant placement is planned, there may be insufficient time for bone formation. Conversely, if the objective is correction of a contour defect (e.g., a buccal defect at a missing tooth site) and the majority of the implant is inserted into native bone for osseointegration, a slowly replaced material will presumably provide long-term space maintenance.

β-TCP is probably best known for its rapid resorption [43]. Lambert et al. compared the healing of rabbit sinuses augmented with xenograft, BCP, and pure β-TCP [44]. Each material supported the formation of new bone, but the bone architecture differed among materials. At 2 months after augmentation, the xenograft had formed an intimate bone bridge between the particles, while the β-TCP graft showed no bone formation. At 6 months after augmentation, there was nothing left in the β-TCP graft. These findings implied more rapid resorption of pure phase β-TCP compared to xenograft and BCP. In another study, Jensen et al. created defects in the mandibles of mini-pigs and grafted them with either autograft, xenograft, or β-TCP; they then harvested bone sections after 1, 2, 4, or 8 weeks [45]. Consistent with the results of other studies, they found that autografts and β-TCP produced slightly more new bone during initial healing (after 4 weeks).

BCP is a combination of two alloplastic materials, generally β-TCP and HA, with ratios adjusted to potentially manipulate their biomedical properties. Cordaro et al. carried out a randomized controlled trial comparing bone healing in grafted human sinuses with either BCP or xenograft at 6 to 8 months after engraftment [43]. The materials differed during later healing, such that less residual synthetic material remained, compared with xenograft material (26.6%). Mahesh et al. grafted human sockets with BGs, then compared bone formation with that achieved using xenografts. Significantly more new bone formed from the BG putty (36–57%) between 4 and 6 months after engraftment. Furthermore, the BG resorbed at approximately 20% per month [46]. Unlike the slower resorbing CP compounds, CS compounds resorb relatively quickly, generally within 8 weeks and certainly by 6 months after engraftment [47].

### 4.2. Similarities and Differences in the Distributions of Alloplastic Bone Graft Products in Multiple Countries

The characteristics of alloplastic bone graft products in each country are shown in Figure 3. In summary, in Japan, there have been relatively few alloplastic bone substitutes approved by the PMDA. These products mainly consist of HA and β-TCP; none consist of CS, CP, or BGs. Recently, carbonate apatite (CA) and octacalcium phosphate (OCP) have been approved. Notably, human bone is carbonate apatite that contains 6–9% carbonate mass in its apatite structure. A previous study revealed that CA could upregulate osteoblast differentiation and was resorbed by osteoclasts [48,49]. OCP is a material that can be converted to HA in physiological conditions and is considered a mineral precursor to bone apatite crystals [50]. The performance of OCP as a bone substitute differs from that of HA materials in terms of its osteoconductivity and biodegradability. OCP elicits a cellular phagocytic response through osteoclast-like cells, similar to that elicited by the biodegradable material β-TCP [51,52,53]. Thus, CA and OCP may be promising alloplastic bone substitutes. Because of the strictness of PMDA approval, there are few other dental bone graft materials approved for periodontal and bone regeneration in Japan (two xenogenic bone graft and no allogeneic bone graft products), excluding bone graft products indicated for maxillofacial and orthopedic uses. Notably, allogeneic bone graft products are also regulated in most countries in Europe. 

Only four kinds of alloplastic bone graft products are approved by both the PMDA and the FDA: Cytrance Granules (GC), OSferion Dental (Olympus Terumobiomaterial), Cerasorb M (Zimmer Biomet Dental G.K.), and Arrow Bone-β-Dental (Brain Base Corporation). Only one alloplastic bone graft product is approved by both the PMDA and the MOHW: Cerasorb M (Zimmer Biomet Dental G.K.). The products available in Japan are indicated mainly for periodontal defects, while only four products are indicated for GBR: Apaceram-AX-Dental (HOYA Technosurgical) for ridge preservation, Neobone (CoorsTek KK) for mineral bone augmentation, and Cytrance Granules (GC) and Bonarc (Toyobo) for GBR. Three of the four products were approved after 2019. Implant treatments after GBR included the sinus lift procedure are also widely performed by Japanese dentists under a self-pay care fee structure, based on clinical evidence and patient consent, using bone graft materials such as allogeneic and xenogenic bone grafts that are not approved by PMDA for off-label use. There were no products approved for the treatment of bone defects derived from peri-implantitis.

Approximately sixfold more alloplastic bone substitute products are approved by the FDA, compared with those approved by the PMDA, despite a previous report that allogeneic bone graft products comprise the major bone graft materials used in the United States. As in Japan, alloplastic bone substitute products approved by the FDA mainly consist of HA and β-TCP, as well as BCP; a few products consist of CS, CP, and BGs. Most alloplastic bone substitute products are indicated for periodontal defects, as well as ridge augmentation, ridge preservation, and sinus lift. Furthermore, 17 products have also been approved for treatment of bone defects derived from peri-implantitis.

A previous study showed that 28 alloplastic bone substitute products were approved by the Korean MHOW through 2019: four HA, eight β-TCP, 15 BCP, and one CP [54]. Approximately four-fold more alloplastic bone substitute products have been approved by the MOHW, compared with those approved by the PMDA. In contrast to Japan, alloplastic bone substitute products approved by the MOHW mainly consist of BCP; none consist of CS or BGs. Most alloplastic bone substitute products are indicated for periodontal defects, as well as ridge augmentation, ridge preservation, and sinus lift. Furthermore, two products have been approved for bone defects derived from peri-implantitis. Only one product, Cerasorb M (Zimmer Biomet Dental G.K.) is approved in both Korea and Japan. Seven alloplastic bone graft products are approved by both the MHOW and the FDA: Cerasorb M (Zimmer Biomet Dental G.K.); MDCP and MDCP Plus (Biometlante); Osteon, Osteon II, and Osteon III (GENOSS); and TCP Dental (Kasios SAS). Compared with Japan and the United States, alloplastic bone graft substitute products that consist of BCP are the main such products in Korea.

Most studies involving clinical randomized controlled trials and split-mouth studies have used similar products: NovaBone (Jacksonville, FL, USA), Curasan (Research Triangle Park, NC, USA), or Biomet (3i) [55]. However, it is difficult to directly compare the product distribution with respect to indications because the descriptions of indications are not standardized among products; there is considerable ambiguity and inconsistency among products in Japan, the United States and Korea. The number of approved products varies among countries and products manufactured by companies tend to be most commonly used in their home countries.

### 4.3. Alloplastic Bone Graft Products for Periodontal and Bone Regeneration

In the context of periodontal regeneration, bone graft materials are required to increase space in patients with non-contained defects such as one-wall defects and class II furcation involvement [56,57]. Preferably, alloplastic bone substitutes will be completely resorbed. A previous study showed that non-resorbable products such as HA sintered at high temperatures tended not to be used for periodontal regeneration because of concerns that residual bone graft materials may cause long-term inhibition of periodontal tissue formation and weak resistance due to re-infection [22,26,27]. For complete bone substitute resorption, 3–6 months is an appropriate interval considering the speed of bone remodeling and creation of space [58,59,60]. In contrast, materials with slow resorption rates are required in situations involving GBR and sinus lift where robust space creation and primary implant stability are needed [59]. Although autologous bone is generally considered the gold standard, single-use autologous bone is not appropriate for GBR because of its high resorption rate [61]. Selection of a product with a suitable resorption rate is necessary for each clinical situation. We also emphasize that an appropriate surgical procedure should be considered in clinical situations. This procedure may include the concomitant use of alloplastic bone substitutes with growth factors, or the use of alternative surgical techniques such as onlay block grafting and distraction osteogenesis [62].

### 4.4. Available Forms of Alloplastic Bone Graft Products

The available forms of bone graft materials are mostly particles in the United States, Japan, and Korea. This trend may be changing in Japan. Since 2019, two products—ReFit Dental (HOYA Technosurgical) and Bonarc (Toyobo)—have been approved in sponge, disk, and rod forms to facilitate operability and handling. Materials with these forms are easy to trim to a size suitable for bone defect management and there is no need cause for concern regarding particles scattered around the defect. These products can also be fixed and sutured at the intended position. Furthermore, the efficiency of β-TCP coated with poly lactide-co-glycolide (β-TCP/PLGA) (Easy graft, Sunstar Inc.) has been demonstrated; this product can solidify after it fills in a bone defect, while retaining its shape. The moldable β-TCP/PLGA graft was effective for ridge preservation, while minimizing both linear and volumetric changes after tooth extraction in sockets with buccal bone deficiency in a dog model [63]. The second major available forms of products were putty in the United States and injection in Korea. The forms of alloplastic substitutes are determined by their chemical components and manufacturing methods. With further development of digital dentistry, alloplastic bone substitutes may be manufactured with forms completely fitted to bone defects before surgery [64,65,66,67]. Currently, customized alloplastic block bones are made using computer-aided design and computer-aided manufacturing or three-dimensional (3D) printing. The advantages of 3D printing include reduced material waste, enhanced optimizable surfaces and porous structures, and shorter operation time. Thus, there is great demand for 3D printing technology; many studies have been published concerning 3D printing technology. Although the evidence regarding 3D-printed alloplastic block bone grafts for ridge augmentation is currently limited to animal studies, the concept is very promising [68,69,70].

### 4.5. Recent Progress in Alloplastic Bone Graft Products

In contrast to allogeneic bone, alloplastic bone substitutes only have the ability to support osteoconduction; their regenerative abilities might generally be weak. Multiple observational studies have provided consistent histological evidence that autogenous and demineralized allogeneic bone grafts support the formation of new attachments. Limited data also suggest that xenogenic bone grafts can support the formation of a new attachment apparatus. In contrast, nearly all available data indicate that alloplastic grafts support periodontal repair, rather than regeneration [71]. Previous studies have shown that particles of alloplastic bone substitutes could be encapsuled by connective tissue during periodontal regeneration [61]. However, recent studies have shown that alloplastic bone graft substitutes composed of BCP (90:10 ratio of β-TCP and HA (Vivoss, Straumann AG; currently Osopia, Regedent)) have the potential to induce ectopic bone formation similar to demineralized freeze-dried bone allograft. This synthetic was compared with xenografts, autografts, and allografts; all were investigated for their abilities to form ectopic bone in rat muscle [72,73]. The same working group found that the synthetic BCP consistently formed ectopic bone in the calf muscles of beagle dogs within 8 weeks after engraftment. This in vivo ectopic bone model demonstrated that while xenografts were not osteoinductive and autogenous bone grafts were resorbed quickly in vivo, ectopic bone formation was reported in demineralized freeze-dried bone allograft and in synthetic BCP grafts [74]. The results from this study indicate that synthetic bone grafts serve as a 3D scaffold and can promote osteoinduction.

Ideal alloplastic bone substitutes demonstrate behavior similar to that of autologous bone with respect to osteoinduction and osteogenesis. Therefore, the concomitant use of alloplastic bone substitutes and growth factors or cell transplantation has been performed to promote periodontal and bone regeneration [74,75]. Depending on the chemical properties of the products involved, growth factors could be released in a controlled manner during decomposition of specific components [76]. In other studies, electrically polarized materials such as HA and β-TCP have been shown to accelerate new bone formation [77,78]. Some ceramics can be ionically polarized by thermoelectrical treatments. The resulting polarized ceramics harbor large, persistent induced electrostatic charges on their surfaces [79]. The surface charges of electrically polarized HA were demonstrated to enhance osteoconductivity, presumably through protein adsorption, as well as cell adhesion, manipulation, and differentiation [80,81,82]. Previous studies reported that the combined application of periodontal ligament-derived mesenchymal stromal cell sheets and β-TCP revealed considerable potential for periodontal regeneration [74,75,83,84,85,86,87]. Overall, the progress in alloplastic bone substitutes has been remarkable thus far. Because the ideal forms and resorption rates of alloplastic bone substitutes differ among patients and clinical situations, dental clinicians should carefully consider the compositions, porosities, and properties of the available products. A sufficient understanding of the properties of alloplastic bone graft products aids in their appropriate selection. Moreover, further studies concerning alloplastic materials are expected to enhance the use of osteoconduction in cell migration and angiogenesis, thereby creating appropriate space without inhibition of wound healing, while maintaining stable blood clot formation and a suitable resorption rate.

Currently, allogeneic and xenogenic bone graft products are popular in both periodontal and bone regeneration applications in the United States [57]. However, there are many advantages of alloplastic bone graft substitutes, such as the absence of potential infectious disease transmission, as well as the absence of ethical or religious controversies. When alloplastic bone substitutes can be used concomitantly with growth factors and/or cell transplantation, there is a reasonably expectation for osteoconduction, osteoinduction, and osteogenesis. Thus, alloplastic bone substitute products may be the first choice for periodontal and bone regeneration therapy because of their safety and predictability.

## 5. Conclusions

To the best of our knowledge, this is the first descriptive report in the field of dentistry that attempts to identify all currently available alloplastic bone graft products approved for use in periodontal and bone regeneration in multiple countries, including the United States, Japan, and Korea. Detailed and accurate information concerning alloplastic bone products was available from three countries (i.e., the United States, Japan and Korea); trends and current statuses were identified. However, information concerning alloplastic bone products was unavailable in other countries and regions (i.e., the European Union, China). There is limited information available regarding the effectiveness and safety of alloplastic bone substitutes approved for use in dental practice. Overall, various alloplastic bone products are available, but this review could not show clear usage criteria for alloplastic bone graft products used in periodontal and bone regeneration. However, the comprehensive assessments in this review may greatly help dental clinicians and surgeons to understand the properties and indications of each alloplastic bone product. They may also aid in the selection of products in various clinical situations. Further studies (e.g., well-designed randomized controlled trials) are necessary to evaluate the clinical efficacies of dental alloplastic bone substitutes. Those studies should consider the current limited information and develop clinical evidence and guidelines that can benefit clinicians everywhere.

In the near future, alloplastic bone substitutes with high safety and standardized quality may be the first choice, instead of autologous bone, when they exhibit robust osteoconductive and osteoinductive capabilities. These products may be used because of their easier handling, high moldability form, and adequate resorption rate, as well as their abilities to be used with growth factors and/or cell transplantation.

## Figures and Tables

**Figure 1 materials-14-01096-f001:**
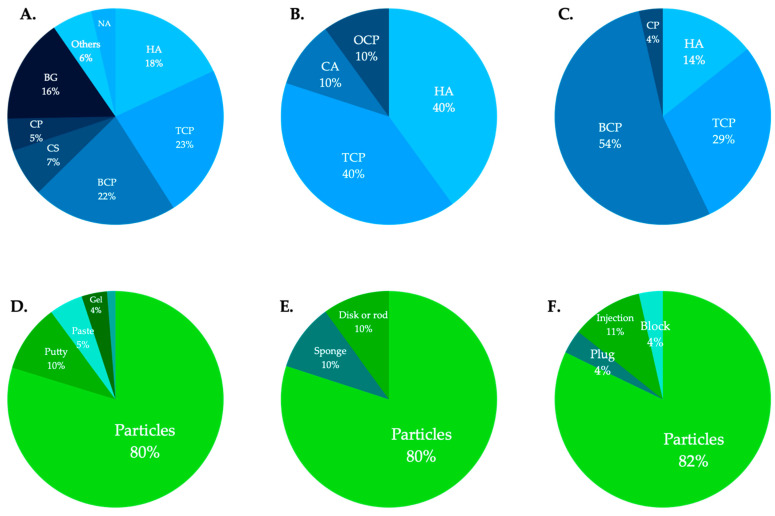
Dental alloplastic bone substitute products according to components and available forms commercially available in the United States (**A**,**D**), Japan (**B**,**E**), and Korea (**C**,**F**). HA = hydroxyapatite, TCP = tricalcium phosphate, BCP = biphasic calcium phosphate, CS = calcium sulfate, CP = calcium phosphate, BG = bioglass, CA = carbonate apatite, OCP = octacalcium phosphate.

**Figure 2 materials-14-01096-f002:**
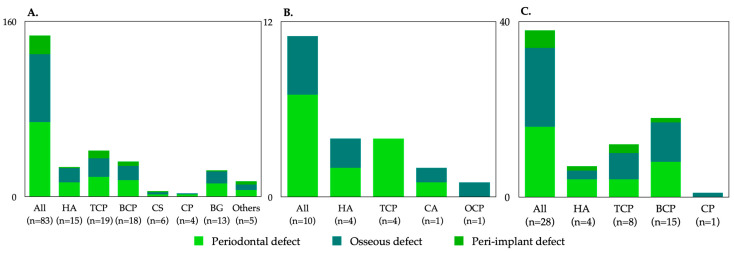
Dental alloplastic bone substitute products with indications commercially available in the United States (**A**), Japan (**B**), and Korea (**C**). HA = hydroxyapatite, TCP = tricalcium phosphate, BCP = biphasic calcium phosphate, CS = calcium sulfate, CP = calcium phosphate, BG = bioglass, CA = carbonate apatite, OCP = octacalcium phosphate.

**Figure 3 materials-14-01096-f003:**
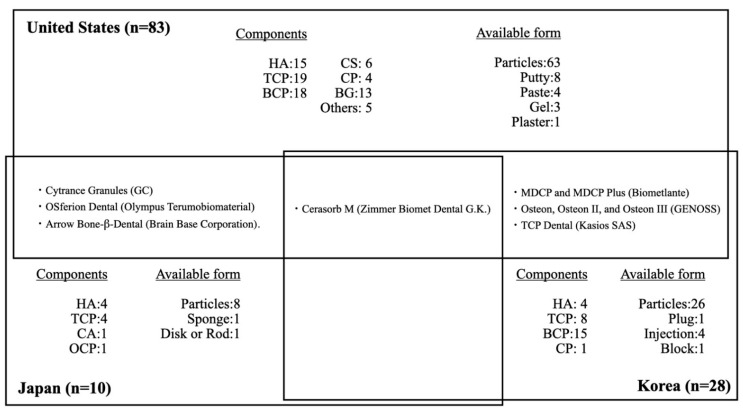
Distribution of dental alloplastic bone substitute products commercially available in the United States, Japan, and Korea. HA = hydroxyapatite, TCP = tricalcium phosphate, BCP = biphasic calcium phosphate, CS = calcium sulfate, CP = calcium phosphate, BG = bioglass, CA = carbonate apatite, OCP = octacalcium phosphate.

**Table 1 materials-14-01096-t001:** Dental alloplastic bone substitute products for periodontal and oral implant applications that have been approved (510(k)) by the Food and Drug Administration (FDA)

Commercial Name	Manufacturer	Compositions	Indication	Available Form	Particle Size or Block Dimensions	Volume/Weight Options	Approval Date
CaSO_4_ (calcium sulfate hemihydrate)	ACE Surgical	Calcium sulfate	Osseous defects	Particles	-	0.5 g, 1.0 g	15 July 2005
Actifuse	Apatech Ltd.	Calcium phosphate	Periodontal, oral, craniomaxillofacial application	Particles	-	-	30 July 2009
Bond Bone	Augma Biomaterials	Calcium sulfate	Osseous defects	Paste	N/A	0.5 cc, 1 cc	17 March 2009
Bond Apatite	Augma Biomaterials	Calcium sulfate (2/3) + HA (1/3)	Periodontal or bony defects	Paste	N/A	-	5 December 2013
TRICOS T	Baxter Healthcare Corp.	BCP + Fibrin matrix	surgically created osseous defects or osseous defects resulting from traumatic injury	Particles	-	-	8 April 2008
TRICOS A	Baxter Healthcare Corp.	Calcium phosphate + fibrin matrix	Surgically created osseous defects, osseous defects resulting from trauma injury	Particles	-	-	6 August 2008
SynthoGraft	Bicon	β-TCP	Traumatic or degenerative multi wall bone defects, augmentation of the sinus floor, augmentation of alveolar ridges, periodontal or other alveolar bone defects and tooth sockets and osteotomies, preservation of the alveolus for preparation of an implant site	Particles	50–500 µm, 500–1000 µm	0.25 g, 0.50 g, 1.0 g, 2.0 g	1 September 2005
BoneGen-TR	Bio-Lok Intl., Inc.	Calcium sulfate + PLLA	Oral surgery: post-extraction Periodontics: intra-osseous defects Endodontics Implantology: dehiscences, fenestrations, sinus lifts	Plaster	Undisclosed	1.5 g	16 May 2006
BonGros HA	BioAlpha Inc.	HA	Augmentation or reconstructive of the alveolar ridge, filling of extraction sockets, elevation of the maxillary sinus floor, filling of peri-implant defects (GBR)	Particles	3000–6000 µm	5 cc, 10 cc, 20 cc, 30 cc	19 May 2009
Fortoss vital bone graft substitute	Biocomposites Ltd.	No description	No description	-	-	-	26 August 2005
Fortoss vital	Biocomposites Ltd.	β-TCP + Hydroxy sulfate	Periodontal defects, maxillofacial defects, dental implant surgery	Paste	N/A	0.5 cc, 1 cc, 2 cc	5 September 2008
Calcium hydroxyapatite implant	Bioform	HA	Periodontal defects, ridge augmentation, extraction sites, craniofacial augmentation, cystic defects	Particles	-	-	27 June 2003
MBCP+	Biomatlante	BCP, HA (20%) + β-TCP (80%)	Periodontal defects, ridge augmentation, extraction sites, craniofacial augmentation, cystic defects	Particles	500–1000 µm	0.5 cc	15 May 2010
MBCP	Biomatlante	BCP, HA (60%) + β-TCP (40%)	Periodontal defects, ridge augmentation, extraction sites, craniofacial augmentation, cystic defects	Particles	500–1000 µm	0.5 cc	16 September 2005
MBCP GEL	Biomatlante	BCP, HA (60%) + β-TCP (40%) + Hydrogel	Periodontal defects, ridge augmentation, extraction sites, craniofacial augmentation, cystic defects	Gel	N/A	0.5 mL, 1 mL, 2.5 mL, 5 mL, 10 mL	2 June 2006
Blue Sky Bio TCP Bone Graft Material	Blue Sky Bio	β-TCP	Periodontal defects, ridge augmentation, extraction sites, sinus lift and sinus floor elevation, cystic defects	Particles	-	-	25 May 2000
Arrowbone-A, Arrowbone-B	Brain Base Corporation	β-TCP	Periodontal defects, ridge augmentation, extraction sites, sinus lift and sinus floor elevation, cystic defects	Particles	250–1000 µm 1000–2000 µm	-	8 December 2009
Caravan Osseolive Dental	Carasan AG	Bioactive calcium-potassium-sodium- phosphate	Periodontal defects (intrabony pockets, bi and tri furcation defects), ridge augmentation, extraction sites, sinus lift and sinus floor elevation, cystic defects	Particles	250–1000 µm 1000–2000 µm	0.5 cc, 1 cc, 2 cc	20 December 2012
Osbone Dental	Carasan AG	HA	Periodontal defects (intrabony pockets, bi- and tri furcation defects), ridge augmentation, extraction sites, sinus lift and sinus floor elevation, cystic defects	Particles	250–1000 µm 1000–2000 µm	0.5 cc, 1 cc	12 January 2011
ReproBone dental grafting material	Ceramisys Ltd.	BCP, HA (60%) + β-TCP (40%)	Periodontal defects, periodontal defects in conjunction with guided tissue regeneration (GTR) and GBR, ridge augmentation, extraction sites, sinus lift and sinus floor elevation, craniofacial augmentation, cystic defects, filling of peri-implant defects (GBR)	Particles	500–1000 µm 800–1500 µm	-	3 November 2011
SynOss synthetic bone graft material (SynOss Granule)	Collagen Matrix Inc.	Carbonate apatite	Periodontal defects, periodontal defects in conjunction with GTR and GBR, ridge augmentation, extraction sites, sinus lift and sinus floor elevation, craniofacial augmentation, cystic defects, filling of peri-implant defects (GBR)	Particles	350–1000 µm	0.5 cc, 1.0 cc, 2.0 cc, 3.5 cc	18 October 2007
SynOss collagen synthetic material (SynOss Putty)	Collagen Matrix Inc.	Carbonate apatite + type 1 collagen	Periodontal defects, periodontal defects in conjunction with GTR and GBR, ridge augmentation, extraction sites, sinus lift and sinus floor elevation, craniofacial augmentation, cystic defects, filling of peri-implant defects (GBR)	Putty	9000 × 8000 µm 11,000 × 10,500 µm 11,000 × 21,000 µm	0.5 cc, 1 cc, 2 cc	12 February 2009
Periophil Biphasic	Cytophil Inc.	BCP, HA (60%) + β-TCP (40%)	Periodontal or oral/maxillofacial defects	Particles	250–500 µm 250–1000 µm 500–1000 µm 1000–2000 µm	0.5 cc, 3.0 cc	18 December 2009
Periophil β-TCP	Cytophil Inc.	β-TCP	Periodontal or oral/maxillofacial defects	Particles	250–500 µm 250–1000 µm 500–1000 µm 1000–2000 µm	0.5 cc, 3.0 cc	22 April 2010
Easy-graft	Sunstar Inc.	β-TCP + PLGA	Extraction defects, periodontal defects, peri-implant defects, GBR, sinus floor augmentation	Putty	500–630 µm 500–1000 µm	0.15 mL, 0.25 mL, 0.4 mL	27 September 2013
Osteograf/D-300	Dentsply	HA	Intrabony periodontal defects, augmentation of bony defects in the alveolar ridge and filling of extraction site	Particles	250–420 µm	-	8 August 2007
Osteograf/D-700	Dentsply	HA	Intrabony periodontal defects, augmentation of bony defects in the alveolar ridge and filling of extraction site	Particles	420–1000 µm	-	6 August 2007
Osteograf/LD-300	Dentsply	HA	Intrabony periodontal defects, augmentation of bony defects in the alveolar ridge and filling of extraction site	Particles	250–420 µm	1.0 g, 5.0 g	6 August 2007
Healos dental bone graft substitute	Depuy Spine	HA + Type 1 bovine collagen	Periodontal or bony defects	Putty	-	-	25 May 2010
Healos II dental bone graft substitute	Depuy Spine	HA + Type 1 bovine collagen	Periodontal or bony defects	Putty	-	-	29 August 2008
CarriGen (Calcigen S)	Etex	Calcium sulfate	Periodontal intrabony defects, alveolar ridge augmentation, extraction sites, sinus lift, cystic defects, craniofacial augmentation,	Paste	N/A	1.5 g	21 December 2010
Frios Algipore	Friadent GMBH	HA	Intrabony defects, ridge augmentation, sinus lift, extraction socket	Particles	-	-	5 Februaly 2003
Cytrans Granules	GC America	Carbonate apatite	Ridge augmentation; periodontal defects; defects after root resection, apicoectomy, cystectomy; extraction sockets; sinus floor elevation; periodontal defects in conjunction with products intended for GTR and GBR; peri-implant defects in conjunction with products intended for GBR	Particles	S: 300–600 µm M: 600–1000 µm	-	17 August 2020
Osteon	Genoss	BCP, HA + β-TCP	Periodontal intrabony defects, alveolar ridge augmentation, extraction sites, sinus lift, cystic defects	Particles	300–500 µm 500–1000 µm 1000–2000 µm	0.25 cc, 0.5 cc, 1.0 cc	24 April 2007
Osteon III	Genoss	BCP, HA (60%) + β-TCP (40%)	Periodontal intrabony defects, alveolar ridge augmentation, extraction sites, sinus lift, cystic defects	Particles	-	-	14 September 2016
Osteon II	Genoss	BCP, HA (70%) + β-TCP (30%)	Periodontal intrabony defects, alveolar ridge augmentation, extraction sites, sinus lift, cystic defects	Particles	-	-	17 January 2012
Osteon Sinus	Genoss	BCP, HA (70%), + β-TCP (30%)	No description	Particles	500–1000 µm 1000–2000 µm	0.5 cc	-
Osteon Lifting	Genoss	BCP, HA (70%), + β-TCP (30%)	No description	Particles	300–500 µm 500–1000 µm	0.5 cc	-
BioActys Granules	Graftys	BCP, HA (60%), + β-TCP (40%)	Periodontal or bony defects	Particles	500–1000 µm 1000–2000 µm	0.5 cc, 1 cc, 2 cc, 5 cc	24 February 2009
Ostim	Herafus Kulzer	HA	Periodontal intrabony defects, alveolar ridge augmentation, extraction sites, sinus lift, cystic defects	Particles	-	-	-
OsteoGen SBRG	Impladent	HA	Periodontal intrabony defects, alveolar ridge augmentation, extraction sites, sinus lift, peri-implant defects	Particles	-	-	27 April 2004
Inion BioRestore	Inion Oy	Bioglass	Periodontal intrabony defects, alveolar ridge augmentation, extraction sites, sinus lift, cystic defects, craniofacial augmentation	Particles	-	-	24 July 2007
ReOss Powder	Intra-lock	HA (50%) + PLGA (50%)	Intraoral/maxillofacial osseous defects, intrabony and furcations periodontal defects, ridge augmentation, extraction sites, sinus elevation	Particles	500–1000 µm	0.5 cc, 1 cc	27 May 2009
ReOss Putty	Intra-lock	HA (50%) + PLGA (50%)	Intraoral/maxillofacial osseous defects, intrabony and furcations periodontal defects, ridge augmentation, extraction sites, sinus elevation	Putty	-	0.5 cc, 1 cc	27 May 2009
ReOss Injectable Gel	Intra-lock	HA (50%) + PLGA (50%)	Intraoral/maxillofacial osseous defects, intrabony and furcations periodontal defects, ridge augmentation, extraction sites, sinus elevation	Gel	N/A	-	27 May 2009
PolyboneDental	Kyungwon Medical	β-TCP	Periodontal intrabony defects, alveolar ridge augmentation, extraction sites, sinus lift, cystic defects, craniofacial augmentation	Particles	200–500 µm	-	16 June 2010
Ceros TCP Granules	Mathys Ltd.	β-TCP	GBR, filling defects after explantation of dental implants, intrabony and bi- and tri- furcations, preparation of implant bed (sinus lift), filling bone defects around dental implant after immediate placement into extraction sockets	Particles	100–500 µm	-	15 September 2010
Mastergraft Resorbable Ceramic Granules	Medtronic	BCP, HA (15%) + β-TCP (85%)	Periodontal defects, alveolar ridge augmentation, extraction sites, sinus lift, cystic defects, craniofacial augmentation	Particles	500 µm	-	9 January 2009
Mastergraft Putty	Medtronic	BCP, HA (15%) + β-TCP (85%) + type 1 bovine bone collagen	Periodontal defects, alveolar ridge augmentation, extraction sites, xinus lift, cystic defects, craniofacial augmentation	Putty	-	-	17 September 2008
Medtronic Dental TCP	Medtronic	β-TCP	Ridge augmentation, sinus augmentation, filling extraction sites, filling of lesions of periodontal origin	Particles	-	-	30 December 2009
Bone Plus BCP	Megagen Implant	BCP, HA (60%) + β-TCP (40%)	Periodontal defects, periodontal defects in conjunction with GTR and GBR, ridge augmentation, extraction sites, sinus lift and sinus floor elevation, cystic defects, filling of peri-implant defects (GBR)	Particles	-	-	2 July 2010
Bone Plus BCP Eagle Eye	Megagen Implant	BCP, HA (60%) + β-TCP (40%)	Periodontal defects, periodontal defects in conjunction with GTR and GBR, ridge augmentation, extraction sites, sinus lift and sinus floor elevation, cystic defects, filling of peri-implant defects (GBR)	Particles	-	-	21 August 2012
Bonemedik-DM	Meta Biomed	BCP, silicon-substituted HA (60%) + β-TCP (40%)	Periodontal defects, periodontal defects in conjunction with GTR and GBR, ridge augmentation, extraction sites, sinus lift and sinus floor elevation, cystic defects, filling of peri-implant defects (GBR)	Particles	-	-	3 June 2008
PerioGlas Bioglass	NovaBone Products	Bioglass	Periodontal defects, alveolar ridge augmentation, extraction sites, sinus lift, cystic defects, craniofacial augmentation	Particles	-	-	1 March 2004
PerioGlas bone graft	NovaBone Products	Bioglass	Periodontal defects, alveolar ridge augmentation, extraction sites, sinus lift, cystic defects, craniofacial augmentation	Particles	90–710 µm	-	14 February 2006
Novanbone Dental Morsels	NovaBone Products	Bioglass	Periodontal defects, alveolar ridge augmentation, extraction sites, sinus lift, cystic defects, craniofacial augmentation	Particles	-	-	16 December 2011
Novabone BBG	NovaBone Products	Bioglass	Periodontal defects, alveolar ridge augmentation, extraction sites, sinus lift, cystic defects, craniofacial augmentation	Particles	-	-	25 March 2000
NovaBone Dental Putty	NovaBone Products	Bioglass, calcium phosphosilicate polyethylene glycol + glycerin, binder	Periodontal defects, alveolar ridge augmentation, extraction sites, sinus lift, cystic defects, craniofacial augmentation	Particles and water soluble binder	32–125 µm 90–710 µm	-	12 February 2007
PerioGlas Plus	NovaBone Products	Bioglass + calcium sulfate binder	Periodontal defects, alveolar ridge augmentation, extraction sites, sinus lift, cystic defects, craniofacial augmentation	Particles	-	-	5 November 2003
PerioGlas Putty	NovaBone Products	Bioglass + gelatin	Periodontal defects, alveolar ridge augmentation, extraction sites, sinus lift, cystic defects, craniofacial augmentation	Putty	-	-	22 February 2007
BBP Bone substitute	OCT.USA	N/A	Ridge augmentation, periodontal defects, extraction sockets	-	-	-	17 June 2004
Osferion D	Olympus Terumo Biomaterials Corporation	β-TCP	Periodontal defects, periodontal defects in conjunction with GTR and GBR, ridge augmentation, extraction sites, sinus lift and sinus floor elevation, cystic defects, filling of peri-implant defects (GBR)	Particles	S: 150–500 µm M: 500–1000 µm L: 1000–2000 µm	-	28 July 2008
Bioresorb macro pore	Oraltronics Dental Implant Technology	β-TCP	GTR, sinus lift, ridge preservation, ridge augmentation	Particles	-	-	15 July 2005
Nanogen	Orthogen	Calcium sulfate	Alone in bone regeneration; mixed with other bone grafts; providing a reservable barrier over other bone grafts	Particles	400–850 µm	-	6 May 2011
Vitomatrix	Orthovita	β-TCP	Periodontal defects, periodontal defects in conjunction with GTR and GBR, ridge augmentation, extraction sites, sinus lift and sinus floor elevation, cystic defects, filling of peri-implant defects (GBR)	Particles	250–1000 µm 1000–2000 µm	-	27 September 2010
Ossaplast Dental	Ossacur	β-TCP	Periodontal defects, periodontal defects in conjunction with GTR and GBR, ridge augmentation, extraction sites, sinus lift and sinus floor elevation, cystic defects, filling of peri-implant defects (GBR)	Particles	500–1000 µm	-	21 February 2006
Cerasorb Dental, Cerasorb M Dental, Cerasorb Perio	Riemser Arzeimittel AG	β-TCP	Periodontal defects, periodontal defects in conjunction with GTR and GBR, ridge augmentation, extraction sites, sinus lift and sinus floor elevation, cystic defects, filling of peri-implant defects (GBR)	Particles	150–500 µm 500–1000 µm 1000–2000 µm	-	20 September 2012
RTR syringe	Septodont	β-TCP	Extraction sockets	Particles	500–1000 µm	-	11 May 2007
Regen Biocement	Steiner	N/A	Bone graft in maxillofacial region	Particles	-	-	28 November 2006
SocketGraft	Steiner	β-TCP	Dental extraction sockets with all walls remaining	Putty	-	-	9 June 2006
Ossoconduct	Steiner	β-TCP	Periodontal defects, alveolar ridge augmentation, extraction sites, sinus lift, cystic defects, craniofacial augmentation	Particles	(Perio) 250–500 µm (Standard) 500–1000 µm (Macro) 1000–2000 µm (Micron) Powder	-	26 October 2010
Straumann BoneCeramic	Strauman	BCP, HA (60%) + β-TCP (40%)	Intrabony periodontal osseous and furcation defects, augmentation of bony defects of the alveolar ridge, filling tooth extraction sites, sinus elevation grafting	Particles	100–500 µm	-	24 September 2020
Chronos-beta-TCP	Synthes	β-TCP	Periodontal defects, alveolar ridge augmentation, extraction sites, sinus lift, cystic defects, craniofacial augmentation	Particles	100–500 µm	-	23 January 2006
Nanogel	Teknimed	Calcium phosphates (30%) + water (70%)	Filling after surgical curettage; bone defects caused by a traumatic lesion on the bone treatment of tuberosity defects, alveolar walls	Gel	0.1–0.2 µm	-	4 March 2009
Odoncer	Teknimed	β-TCP	Periodontal defects, periodontal defects in conjunction with GTR and GBR, ridge augmentation, extraction sites, sinus lift and sinus floor elevation, cystic defects, filling of peri-implant defects (GBR)	Particles	-	-	16 April 2007
Osteocaf	Texas Innovative Medical Devices	HA (12%) + Calcium phosphates (66%) + PLGA (22%)	Intraoral/maxillofacial osseous defects, intrabony and furcations periodontal defects, ridge augmentation, extraction sites, sinus elevation	Particles	250–1200 µm	-	20 April 2011
C-Graft	The Clinician’s Preference	Calcium phosphate	Periodontal intrabony defects, alveolar ridge augmentation, extraction sites, sinus lift	Particles	300–1000 µm	-	10 December 2003
ShefaBone SCPC	The Implantech	Bioglass	Periodontal intrabony defects, alveolar ridge augmentation, extraction sites, sinus lift, cystic defects, craniofacial augmentation	Particles	-	-	14 July 2016
Theriridge block	Therics	HA	Augmentation of deficient maxillary and mandibular alveolar ridges	Particles	12 µm	-	26 March 2003
CALC-I-OSS	Ultradent	β-TCP	Intraoral/maxillofacial osseous defects, intrabony and furcations periodontal defects, ridge augmentation, extraction sites, sinus elevation	Particles	315–1600 µm	-	19 July 2005
Unigraft	Unicare	Bioglass	Periodontal defects, periodontal defects in conjunction with GTR and GBR, ridge augmentation, extraction sites, sinus lift and sinus floor elevation, cystic defects, filling of peri-implant defects (GBR)	Particles	200–400 µm 200–600 µm	-	27 January 2000
Ossiform	Unicare	Bioglass	Periodontal intrabony defects, alveolar ridge augmentation, extraction sites, sinus lift, cystic defects, craniofacial augmentation	Particles	-	-	7 May 2002
Bonalive	Vivoxid	Bioglass SiO_2_ (53%) + Na_2_O (23%) + CaO (20%) + P20 5 (4%)	Periodontal or bony defects	Particles	-	-	25 June 2007

**Table 2 materials-14-01096-t002:** Dental alloplastic bone substitute products for periodontal and oral implant applications approved by the Japanese Pharmaceuticals and Medical Devices Agency. ○; Approval. ×; Unapproved

Commercial Name	Manufacturer	Compositions	Porosity	Available Form	Particle Size	Compressive Strength (MPa)	Resorption Rate	Periodontal Application	Implant Application	Approved Date
Apaceram-AX-Dental	HOYA Technosurgical	HA	82.5 ± 5.5%	Particles	600–1000 μm 1000–2000 μm	0.7	Non-resorbable	×	○ (ridge preservation)	2019.08
Neobone	CoorsTek KK	HA	72–78%	Particles	500–1000 μm 1000–2000 μm	12–18	Non-resorbable	×	○ (mineral bone augmentation)	2003.06
Bonetite	HOYA Technosurgical/Morita	HA	Dense	Particles	Perio: 300–500 μm Standard: 500–1000 μm	Undisclosed	Non-resorbable	○	×	1985.12
ReFit Dental	Hoya/Kyocera	HA (80%) + type I Collagen(20%)	92−98% 100–500 µm pore diameter	Sponge	-	Undisclosed	Resorbable	○	×	2019.09
Cytrans Granules	GC	Carbonate apatite	Undisclosed	Particles	S: 300–600 μm M: 600–1000 μm	Undisclosed	1–2 years	○	○	2017.12
OSferion Dental Terufill Osfill	Olympus Terumobiomaterial/Morita/Kyocera	β-TCP	77.5 ± 4.5%	Particles	S: 150–500 μm M: 500–1000 μm L: 1000–2000 μm	0.9	16 weeks	○	×	2015.11
Cerasorb M	Curasan AG/Zimmer Biomet Dental G.K.	β-TCP	65%	Particles	S: 150–500 μm M: 500–1000 μm L: 1000–2000 μm	Undisclosed	6–12 months	○	×	2012.01
Arrow Bone-β-Dental	BrainBase Corporation	β-TCP	75%	Particles	AG1: 250–1000 μm AG2: 1000–2000 μm	> 1.0	16 weeks	○	×	2013.12
Palton Pearl Bone	CatalyMedic Engineering	β-TCP	73–82%	Particles	150–500 μm 500–1000 μm	Undisclosed	Undisclosed	○	×	2017.12
Bonarc	Toyobo	Octa calcium phosphate (OCP)	Undisclosed	Disk or rod	Disk: 9 mm diameter, 1.5 mm width Rod: 9 mm diameter, 10 mm width	Undisclosed	Undisclosed	×	○	2019.05

**Table 3 materials-14-01096-t003:** Dental alloplastic bone substitute products for periodontal and oral implant applications approved by the Korean Health Insurance Review and Assessment Service

Product Name	Manufacturer	Composition	Indication	Available Form	Morphology
OssaBase-HA	Lasak	HA	Remodeling of the alveolar ridge, treatment of periodontal defects, treatment of bone defects around dental implants, sinus lift, filling of bone defects after surgical extractions to prevent alveolar atrophy, filling of bone defects after extirpation of cysts	Particles	Macro-nano bone-like structure with 83% interconnected porosity
Ovis Bone HA	Dentis	HA	Periodontal bone defects, intrabony defects, extraction sites, ridge augmentation, sinus lift, cystic cavities	Particles	Well-formed macro/micro-porous porosity
CollaOss (Block), Ossbone Collagen	SK Bioland	HA (90 ± 5%) + collagen (10 ± 5%)	Periodontal bone defects, intrabony defects	Plug	Well-formed macro/micro-porous porosity
CollaOss (Putty)	SK Bioland	HA (90 ± 5%) + collagen (10 ± 5%)	Periodontal bone defects, intrabony defects	Particles	Well-formed macro/micro-porous porosity
CollaOss (Syringe)	SK Bioland	HA (90 ± 5%) + collagen (10 ± 5%)	Periodontal bone defects, intrabony defects	Putty	-
DualPor Collagen D-Putty	OssGen	HA (60%) + bovine atelo, collagen (0.3%) + distilled water (39.7%)	No description	-	-
DualPor Collagen D-Injection	OssGen	HA (60%) + bovine atelo, collagen (0.3%) + distilled water (39.7%)	No description	Block	-
BoneSigma TCP	SigmaGraft	β-TCP 100%	Extraction sockets, horizontal and vertical augmentation, peri-implant defects, periodontal regeneration, ridge augmentation, sinus floor elevation	Particles	>95% β-TCP, interconnected macro and micro porous structure
Excelos Inject	BioAlpha	β-TCP etc.	Sinus floor elevation, alveolar bone augmentation, extraction socket preservation	Injectable	β-TCP particle (size: 45–75 μm) with hydrogel (poloxamer and hydroxypropyl methylcellulose)
Excelos (TCPGLD)	BioAlpha	β-TCP 100%	Sinus floor elevation, alveolar bone augmentation, extraction socket preservation	Particles	100% β-TCP, average 80% macro-porosity (pore size: 100–300 μm)
Excelos (TCPGMD, TCPGLD)	BioAlpha	β-TCP 100%	Sinus floor elevation, alveolar bone augmentation, extraction socket preservation	Particles	-
Mega-TCP (CGL)	CGbio	β-TCP 100%	No description	Particles	Porous structure like human cancellous bone > 99% interconnectivity, average 75% macro-porosity (pore size: 100–300 μm)
Mega-TCP (CGM, CGL)	CGbio	β-TCP 100%	No description	Particles	-
Sorbone	Meta-Biomed	β-TCP 100%	Extraction sockets, cystic cavities, periodontal defects, intrabony defects, ridge augmentation, sinus floor elevation	Particles	Average 55–60% macro-porosity
SynCera	Oscotec	β-TCP	No description	Particles	Macro- and micro-porosity
Cerasorb M	Curasan	β-TCP	No description	Particles	Micro-meso-macro pore (pore size 5–500 μm), approximately 65% porosity with full range of pore sizes and interconnected porosities
Bio-C	Cowellmedi	β-TCP + HA	No description	-	-
Boncel-Os	BioAlpha	β-TCP + HA	Ridge augmentation, extraction sockets, periodontal defects, sinus lift	Particles	High porosity, interconnected porous structure
BoneSigma BCP	SigmaGraft	β-TCP (40%) + HA (60%)	Ridge augmentation, extraction sockets, cystic cavities, sinus floor elevation, periodontal defects, peri-implant defects	Particles	Micro- and macro-porosity
Frabone Dental	Inobone	β-TCP (40% ± 5%) + HA (60% ± 5%)	No description	Particles	150–300 μm macropore, average 8.1 μm micropore, 0.7 mm porous particle size
Frabone Dental Inject	Inobone	β-TCP + HA	No description	Injectable	100–300 μm micropore, 0.7 mm porous particle size
Genesis-BCP	Dio	β-TCP (40%) + HA (60%)	No description	Particles	70% complete interconnected porosity, 75% macropore (300–700 μm), 25% micropore (< 10 μm)
MBCP	Biometlante	β-TCP + HA	Sinus lift augmentation, ridge augmentation, alveolar regeneration, alveolar regeneration, intra-osseous pockets	Particles	-
MBCP Plus	Biometlante	β-TCP + HA	Sinus lift augmentation, ridge augmentation, alveolar regeneration, alveolar regeneration, intra-osseous pockets	Particles	70% porosity with 35% microporosity, 1/3 micropores (< 10 μm) and 2/3 macropores (300–600 μm)
New Bone	Genoss	β-TCP + HA	Ridge augmentation, extraction sites and osteotomy, cystic cavities, sinus lift, periodontal defect	Particles	80% porosity (pore size: 200–400 μm), 0.2–2.0 mm porous particle size
Osspol Dental	Genewel	β-TCP (40%) + HA (60%)	No description	Particles	-
Osteon	Genoss	HA (β-TCP)	Periodontal/infrabony defects, ridge augmentation, extraction sites (implant preparation/placement), sinus lift, cystic cavities	Particles	Interconnected porous structure similar to that of human cancellous bone; 77% porosity (pore size: 300–500 μm), irregular shaped particles: (granule) 0.3–2.0 mm, (sinus, syringe) 0.5–2.0 mm, (lifting, syringe) 0.3–1.0 mm
Osteon II	Genoss	β-TCP + HA	Periodontal/infrabony defects, ridge augmentation, extraction sites (implant preparation/placement), sinus lift, cystic cavities	Particles	Interconnected porous structure similar to that of human cancellous bone; > 70% porosity (pore size: 250 μm), irregular shaped particles: (granule) 0.2–2.0 mm, (sinus, syringe) 0.5–2.0 mm, (lifting, syringe) 0.2–1.0 mm
Osteon III	Genoss	β-TCP + HA	Periodontal/infrabony defects, ridge augmentation, extraction sites (implant preparation/placement), sinus lift, cystic cavities	Particles	Interconnected macro and micro porous structure (< 80% porosity), particle size: (granule) 0.2–2.0 mm, (sinus, syringe) 0.5–2.0 mm, (lifting, syringe) 0.2–1.0 mm, > 70% crystallinity CaP = 1.59
Osteon III Collagen	Genoss	β-TCP + porcine collagen (95%)	Alveolar bone defects	Cylinder	Particle size: 0.2–1.0 mm
Osteon Sinus	Genoss	HA (β-TCP)	Sinus lift	-	-
Ovis Bone BCP	Dentis	β-TCP + HA	Periodontal bone defects, intrabony defects, extraction sites, ridge augmentation, sinus lift, cystic cavities	-	70% porosity (pore size: 20 μm), particle size: 0.3–2.0 mm
TCP Dental	Kasios SAS	β-TCP (95%) + HA (5%)	Sinus graft, bone loss correction, filling alveoli, periodontology	Particles	Interconnected macro-porosity, >90% porosity
Q-OSS+	Osstem Implant	β-TCP (80% ± 5%) + HA (20% ± 5%)	No description	Particles	Porous structure
Topgen-S	Toplan	β-TCP (80% ± 5%) + HA (20% ± 5%)	No description	Particles	Interconnected macro and microporous, particle size: 1.0–2.0 mm
Inno-CaP	Cowellmedi	Calcium phosphate (100%)	Sinus lift, guided bone regeneration	Particles	Particle size: 0.41–1.4 mm

## Data Availability

Data is contained within the article.

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
