# Peer review of "Alloplastic Bone Substitutes for Periodontal and Bone Regeneration in Dentistry: Current Status and Prospects"

_materials, 2021, doi:10.3390/ma14051096_

Round 1

Reviewer 1 Report

The manuscript "Alloplastic bone substitutes for periodontal and bone regeneration in dentistry: current status and prospects" is an interesting review work with regards to modern biomedical materials. In my opinion, the article is a quite straightforward review manuscript and deserves to be considered for publication. There are a few small editorial aspects which could be improved in my opinion - please see commented manuscript attached.

In my opinion, the section Materials and Methods should be renamed to Methodology since this is a review, not an experimental article.

The authors should mention synthesis, advantages and disadvantages of these routes, at least briefly, since the materials of interest here are of synthetic nature.

//Minor Revision

Reviewer 2 Report

Dear Authors please put keyword in alphabetic order. 

This is a very complex review that could lead a less experienced reader to get confused. I would suggest reviewing the introduction section with clearer statements on the use of alloplastic's material.

In the section Conclusions you say: "this is the first descriptive report in the field of dentistry that attempts to identify all currently available alloplastic bone graft products approved for use in periodontal and bone regeneration"

Maybe you should mention some working groups that have worked a lot on this type of materials.

Reviewer 3 Report

The manuscript topic is actual and the paper has merit.

Paper fits within Materials aim and scope.

The topic is really debated in the recent literature and I think it may be attractive, adequate and interesting for the journal readers. However there are some point that authors should address in order to have a final more complete paper. Authors should underline the limitation of the value of the study, and the clinical and surgical implication of the presented study should be added. At this stage the paper seems to be directed to researchers and not surgeons or clinicians. Please emphasize the clinical application of the study, and its scientific rationale.

References are inadequate. Introduction section is poor.

Science is running so some more references about the recent five years (2016-2021) CLINICAL reconstructive option just published have to be added.At the same time discussion is poor.
In the discussion section authors should compare the results of the present study with others one presented and published in the literature.

Please add some samples as the following:
Other important growth factors related to bone substitutes material and clinical studies are the following, please add:

Reviewer 4 Report

the authors provided a complete list of available alloplastic biomaterials in US, Japan, and Korea. the summary is informative however, it is more suitable to be assigned as a report rather than a review. below are some points to be considered.

  1. the abstract lacks the aim of the study and the main reason behind that. it should clearly state the existing problem that the paper is going to address.
  2. Table 3 is less informative compared to other tables. please explain?
  3. It is advised to provide a summary of the discussion section using a well-designed table.
  4. 2nd, 3rd, and 4th paragraphs in the discussion section seem to be unnecessary. many of the points are already available in the tables and others can be summarized in the provided tables.
  5. paragraph 8 of the discussion section needs to be revised. The topic of osteoinductivity of the alloplastic biomaterials is very complicated. appropriate references need to be considered as well.
  6. the discussion section lacks a valuable comparison/contrast of the target biomaterials.
  7. the format of the discussion section can be remarkably improved.

Reviewer 5 Report

The manuscript under review attempts to describe current status and prospects of alloplastic bone substitutes for periodontal and bone regeneration in dentistry.

This paper is well organized and easy to read.

This study will be very helpful to understand alloplastic graft and markets.

I would like to thank to authors about good work.

However, the paper did not include contents about recent 3D printed alloplastic bone. I recommend that the discussion section should include 3D printed alloplastic bone grafts. These references would be helpful to describe about 3D printed alloplastic bone grafts.

1. Kim, J.-W.; Yang, B.-E.; Hong, S.-J.; Choi, H.-G.; Byeon, S.-J.; Lim, H.-K.; Chung, S.-M.; Lee, J.-H.; Byun, S.-H. Bone Regeneration Capability of 3D Printed Ceramic Scaffolds. Int. J. Mol. Sci. 2020, 21, 4837. https://doi.org/10.3390/ijms21144837

2. Spath, S.; Drescher, P.; Seitz, H. Impact of Particle Size of Ceramic Granule Blends on Mechanical Strength and Porosity of 3D Printed Scaffolds. Materials 2015, 8, 4720-4732. https://doi.org/10.3390/ma8084720

3. Lim, H.-K.; Hong, S.-J.; Byeon, S.-J.; Chung, S.-M.; On, S.-W.; Yang, B.-E.; Lee, J.-H.; Byun, S.-H. 3D-Printed Ceramic Bone Scaffolds with Variable Pore Architectures. Int. J. Mol. Sci. 2020, 21, 6942. https://doi.org/10.3390/ijms21186942

I just recommend that typo-grammatical errors should be corrected.

Thank you 

Round 2

Reviewer 4 Report

authors tried to update and revisit some parts within the text. the paper seems to have a better view for publication.

Author Response

Response to Reviewer.4 comments
Thank you very much for your careful reading and kind comments. We have made corrections based on your suggestions. Please let us know if there are still any issues to be addressed.

Point 1: English language and style are fine/minor spell check required 

Response 1: We have carefully checked the whole text in the manuscript and corrected typographical errors. The modified sections are indicated in red font in the revised version. We have asked the English editing service to check the spelling, and presented about it in the acknowledge section.